# Bimetallic Nanomaterials-Based Electrochemical Biosensor Platforms for Clinical Applications

**DOI:** 10.3390/mi13010076

**Published:** 2021-12-31

**Authors:** Palanisamy Kannan, Govindhan Maduraiveeran

**Affiliations:** 1College of Biological, Chemical Sciences and Engineering, Jiaxing University, Jiaxing 314001, China; 2Materials Electrochemistry Laboratory, Department of Chemistry, SRM Institute of Science and Technology, Kattankulathur 603203, Tamil Nadu, India

**Keywords:** bimetallic nanomaterials, gold, silver, platinum, palladium, glucose, real samples, clinical diagnostics

## Abstract

Diabetes is a foremost health issue that results in ~4 million deaths every year and ~170 million people suffering globally. Though there is no treatment for diabetes yet, the blood glucose level of diabetic patients should be checked closely to avoid further problems. Screening glucose in blood has become a vital requirement, and thus the fabrication of advanced and sensitive blood sugar detection methodologies for clinical analysis and individual care. Bimetallic nanoparticles (BMNPs) are nanosized structures that are of rising interest in many clinical applications. Although their fabrication shares characteristics with physicochemical methodologies for the synthesis of corresponding mono-metallic counterparts, they can display several interesting new properties and applications as a significance of the synergetic effect between their two components. These applications can be as diverse as clinical diagnostics, anti-bacterial/anti-cancer treatments or biological imaging analyses, and drug delivery. However, the exploitation of BMNPs in such fields has received a small amount of attention predominantly due to the vital lack of understanding and concerns mainly on the usage of other nanostructured materials, such as stability and bio-degradability over extended-time, ability to form clusters, chemical reactivity, and biocompatibility. In this review article, a close look at bimetallic nanomaterial based glucose biosensing approaches is discussed, concentrating on their clinical applications as detection of glucose in various real sample sources, showing substantial development of their features related to corresponding monometallic counterparts and other existing used nanomaterials for clinical applications.

## 1. Introduction

Diabetes, a clinical chronic condition caused by elevated levels of glucose, occurs due to the malfunctioning of pancreatic β-cells responsible for production of insulin which strictly controls the glucose level in the blood. The elevated level of glucose creates serious and greater life-threatening conditions such as cardiac, nervous, renal, ocular, cerebral and peripheral vascular diseases in diabetic patients. Diabetes is a major health problem causing 4 million deaths each year and ~172 million people suffering from diabetes worldwide. It has been estimated that the number of diabetic patients would be more than twofold of the current patients worldwide [1,2]. Although there is no treatment for diabetes, the blood glucose level of diabetes patient should still be closely observed to avoid further difficulties. Due to the rising anxiety from this disease, constant checking of glucose in blood has become a potential tendency resulting in fabrication of precise and sensitive blood sugar detection devices for clinical diagnosis and personal care. Moreover, the crucial desires of glucose testing devices for environmental and food monitoring, clinical diagnostics and the development of renewable and sustainable energy fueled extensive academic and commercial efforts to develop glucose sensors with exceptional selectivity and sensitivity, good reliability, fast response, and low cost [3,4,5].Owing to the unique structure and dimension dependent physico- and electrochemical properties, metallic nanoparticles or nanostructures have predominantly fascinating various technological uses [6,7,8,9]. These nanostructured metal particles have stimulated abundant attention for significant various biosensing applications because of their facile preparation, ease of surface functionalization and construction of many biosensor platforms [10,11,12,13,14]. However, these metallic nanomaterials play different roles in various electrochemical biosensing systems based on their exclusive properties with their functions of immobilization of biomolecules, catalysis of electrochemical reactions, enhancement of electron transfer process, labeling biomolecules and acting as reactants [15,16,17,18,19,20,21,22]. Interestingly, bimetallic nanoparticles are a promising new class of nanomaterials to meet increasing catalytic and sensor industrial demand for optimized catalysts for several electrochemical reactions [23,24,25]. Bimetallic nanoparticles act as a multifunctional platform because their properties are dependent on the composition, size, and shape, so their synthetic approaches and technological applications have fascinated many researchers [26,27,28,29]. Over the past few years, the motivation is to develop the novel catalytic properties of bimetallic nanomaterials, which can be tuned by their morphology, size, and composition of alloy nanocrystals containing inexpensive and readily available late-3D transition metals have been extensively investigated [24,30,31,32,33]. However, bimetallic nanoparticles are effective in controlling the morphology and composition and the synthetic methods are generally carried out within high melting point organic solvents and with the help of vast quantity of surfactants. The surface adsorption of the long carbon chain hydrocarbons often results in contaminated crystal facets. The surface contamination frequently decreases their active sites and deteriorates the catalytic activity, which hinders their further catalytic or sensing applications [34,35].

Bimetallic nanoparticle based biotechnology is a burgeoning field with immense potential for real world clinical applications. To realize this potential, especially for therapeutic applications, it is necessary to design and engineer nanoparticles that may be specifically targeted to analyte molecule, as well as to explicitly produce longer stability. Thus, these engineered nanomaterials may serve as unique multi-dimensional scaffolds, whose properties can vary significantly from their bulk material counterparts. With remarkable achievements in nanotechnology based biosensor platforms, bimetallic nanomaterial-based electrochemical signal amplifications have great potential for enhancing both sensitivity and selectivity for electrochemical glucose biosensors. Glucose, a vital biomolecule, essentially exists in the human blood that delivers energy for regular biological activity of human beings. However, human beings are physically susceptible to diabetes mellitus with its excess ingestion in the human body, which has been observed as one of the major diseases in the world that subsequently end up into physical disorders or even death [36,37]. As a result, an accurate and sensitive glucose sensor is of potential interest in people’s day-to-day life.

## 2. Bimetallic Alloy Nanomaterials Based Glucose Detection

The development of nanotechnology offers new horizons for the application of nanomaterials in bioelectrochemical analysis [38,39]. Bimetallic alloy nanomaterials are composed of two or more different metals. Bimetallic nanomaterials are considered more effective than monometallic counterparts because of their synergistic characteristics [23,29]. Due to the exclusive structure, it possesses several outstanding properties such as faster electron transportation, a higher specific surface area, and superior biocompatibility, which result in favorable applications in electrochemical biosensors [40]. For instance, Yeo and Johnson developed copper based bimetallic alloys of Ni, Fe and Mn electrode materials for the detection of glucose under alkaline solutions [41]. Mn_5_Cu_95_ exhibited great improvement in sensitivity towards the sensing of glucose.

Mallouk and co-workers developed active alloy-based electrodematerials for the oxidation of glucose, displaying both Pt and Pb active materials. However, a serious defect with the Pt_2_Pb materials is its deprived resistance to the poisoning of chloride ions [42]. The improved electrocatalytic activity of the Pt-Pb nanostructures was further studied towards the detection of glucose without the use of any enzymes by Sheu et al. [43]. One conventional method to produce bimetallic alloys (such as Pt-Pb) is by vacuum arc-melting of two pure metal targets, followed by long-time annealing. The resulting PtPb button electrodes possess shiny and mirror-like surfaces, which are not ideal for the electrochemical detection of glucose [44]. Chen and co-workers published a non-enzymatic amperometric glucose sensor based on PtPb and PtIr materix for the sensitive detection of glucose [45,46]. Their amperometric sensitivities increase in the order of Pt-Pb (0%) < Pt-Pb (30%) < Pt-Pb (70%) < Pt-Pb (50%). Indeed, in the case without chloride ions, there is a shoulder-like peak sitting at around −200 mV, just beside the large current peak at 0 mV. In the presence of chloride ions, the shoulder-like current peak disappears. Secondly, at each glucose concentration, the peak values of the current density at both 0 mV and +300 mV (with Cl ions) were relatively higher than those at − 80 mV and +400 mV (without Cl ions). These phenomena are both considered to be the major influence of chloride ions on the nanoporous Pt-Pb (50%) electrode (Figure 1A,B) [45]. Based on the well-accepted mechanism of electro-oxidation of glucose on Pt electrodes in neutral media, the following hypothesis of a competitive adsorption model was proposed. At potentials applied more positive than that of +300 mV, the adsorbed intermediates are oxidized and formed glucono-lactone or gluconic acid products [45].

Zhao et al., developed a new type of self-supporting and flexible electrode based on graphene paper (GP) aided 3D monolithic nanoporous gold (NPG) framework (NPG/GP), and followed by a layer of highly compact, well-spread, and bimetallic PtCo alloy nanoparticles via a simple and efficient ultrasonic electrodeposition method. There is a synergistic effect from the electrocatalytically active PtCo alloy nanoparticles, the large-active-surface area and highly conductive 3D NPG framework, and the mechanically robust and stable GP electrode substrate, the obtained PtCo alloy nanoparticles modified NPG/GP (PtCo/NPG/GP) displayed higher mechanical strength and stable electrochemical sensing performances toward nonenzymatic detection of glucose with an extensive linear range from 35 µM to 30 mM, and the LOD of 5 µM (S/N = 3) with a high sensitivity of 7.84 µA cm^−2^ mM^−1^. Furthermore, the real sample uses of the fabricated PtCo/NPG/GP was reported for the in-vitro detection of blood glucose in real clinic samples [47].

Shim et al. synthesized Au@Pt core-shell NPs by the use of a sonochemical approach (Figure 1C) [48]. Then, the Au is integrated into nano-channels through an electrochemical method (Au@Pt/Au NPs) as a non-enzymatic detection of glucose (0.5–10.0 μM and 0.01–10.0 mM). The LOD of glucose in PBS saline solution has been determined to be 445.7 (±10.3) nM [48]. Later, Zhang et al., prepared two types of self-supported electrodes with Cu_x_O or Cu_x_O/Ag_2_O (x = 1, 2) nanowires grown on the nanoporous surface via de-alloying of Cu_50-x_Zr_50_Ag_x_ (x = 0 and 7.5 at%) metallic glasses, followed by anodizing and calcination (Figure 1D). The influence of inserting Ag metal into Cu_50_Zr_50_ MG sample on the morphology and electrochemical performance of electrodes were methodically studied. Compared with cluster-like copper monometallic oxide nanowire nanoporous Cu (Cu_x_O@NPC, x = 1, 2) electrodes, the in-situ grown copper-silver bimetallic oxide nanowire on nanoporous Cu-Ag (Cu_x_O/Ag_2_O@NP-CuAg) electrodes with tip convergence and hierarchical porous structure showed a superior electro-oxidation response toward glucose. As prepared Cu_x_O/Ag_2_O@NP-CuAg electrodes exhibited a higher sensitivity of 1.31 mA mM^−^^1^ cm^−^^2^ and a longer linear range up to 15 mM along with the LOD of 0.5 µM (S/N = 3). The obtained improved electro-catalytic response is primarily due to the synergistic effect of Cu and Ag components as well as exclusive structural characteristics [49].

Zhang et al., attempted the facile preparation of Cu/Ni bimetallic nanocatalyst towards electro-oxidation of glucose. Firstly, carboxylated multi-walled carbon nanotubes (CMWCNTs) were chemically grafted on the surface of indium tin oxide (ITO) glass through silanization and amide coupling reactions; secondly, the in-situ facile electrodeposition method offered discrete nucleation sites for Cu/Ni bimetallic catalysts. The prepared Cu/Ni bimetallic catalyst showed ultra-high electrochemical activity; the catalytic current density for glucose oxidation was found to be over 6.7 mA mM^−1^ cm^−2^ (Figure 2). The linear response spanned three orders of magnitude of glucose concentration ranging from 1 µM to 1 mM. Notably, the Ni plays a principal role over Cu in electrocatalytic oxidation of glucose, thus enhancing detection strategy for nonenzymatic glucose sensors. The glucose sensor developed in this work is low cost and ready to use, and has easy bulk synthesis [50]. Recently, Kullavadee and co-workers developed a bifunctional gold–copper alloy nanoparticle (AuCu alloy NPs) catalyst electrode for highly sensitive and selective sensors for non-enzymatic glucose and hydrogen peroxide (H_2_O_2_) detection [51]. A series of AuCu alloy NPs with various metal ratios were obtained via a co-reduction reaction. The morphology of AuCu alloy NPs was transformed from highly branched structures (nanourchin, nanobramble, nanostar, nanocrystal) to a spherical shape by changing Au content during the preparation process. Cu-rich AuCu nanobramble and Au-rich AuCu nanostar exhibited selective electrocatalysis behaviors toward electro-oxidation of glucose and electro-reduction of H_2_O_2_, respectively. The AuCu nanobramble–based sensor possesses significant potential in glucose detection in the linear range of 0.25 to 10 mM and a sensitivity of 339.35 μA mM^−1^ cm^−2^ with an LOD of 16.62 μM, which is an acceptable selectivity and good stability. On the other side, the AuCu nanostar–based electrode showed good electrochemical responses toward H_2_O_2_ reduction with LOD and sensitivity of 10.93 μM and 133.74 μA mM^−1^ cm^−2^, respectively. The displayed good sensing responses derived from the synergistic surface structure and atomic composition effects, which leads AuCu alloys to be a potential nanocatalyst for sensing both glucose and H_2_O_2_ [51].

Deng and co-workers reported the preparation of morphology-controlled Cu-Sn alloy nanosheet arrays supported on carbon fiber paper (CP) substrate (Cu-Sn/CP) via a simple one-step electro-deposition method at room temperature. Getting advantages from the large active surface area, significant ion transport channels, and a robust synergistic catalytic effect between Cu and Sn, the synthesized Cu-Sn/CP functioned as a self-supported electrode for effective non-enzymatic glucose sensing (scheme in Figure 3) [52]. The electrochemical behaviors of 1mM glucose were studied using CP, Cu/CP, Sn/CP, and Cu-Sn/CP electrodes in 0.1 M NaOH at a scan rate of 100 mV s^−1^. As shown in Figure 3A, bare CP electrode and Sn/CP electrode showed insignificant oxidation current density change. However, Cu/CP electrode exhibited an obvious peak at about +0.58 V, which could be attributed to the Cu (III)/(II) redox couple during the electro-oxidation of glucose. Mainly, the Cu-Sn/CP electrode presented the enhanced anodic oxidation peak current density in the presence of 1 mM glucose in 0.1 M NaOH solution. The probable mechanism of glucose oxidation at Cu-Sn/CP electrode is explained by the following equations [52,53]:Cu + 2OH^−^ → CuO (II) + H_2_O + 2e^−^(1)
CuO (II) + H_2_O → Cu(OH)_2_ (II)(2)
Cu(OH)_2_ (II) + OH ^−^ → CuOOH (III) + H_2_O + e^−^(3)
CuOOH (III) + C_6_H_12_O_6_ → Cu(OH)_2_ (II) + C_6_H_10_O_6_ + H_2_O(4)

The LSV curves of Cu-Sn/CP electrodes in various concentrations of glucose (from 1 to 10 mM) in 0.1 M NaOH at 100 mV s^−1^ display an increase of the oxidation peak current signal (Figure 3B), indicating the tremendous electro-catalytic activity of Cu–Sn/CP toward the electro-oxidation of glucose. The electro-chemical kinetics of glucose on the Cu-Sn/CP electrode has also been examined by CVs in 0.1 M NaOH at various scan rates ranging from 10 to 100 mV s^−1^ (Figure 3C). The current densities of both anodic (I_pa_) and cathodic (I_pc_) peaks keep good linearity with the scan rates (Figure 3D), demonstrating that the electro-catalytic reaction of glucose on the Cu-Sn/CP electrode is a surface-controlled process. Under ideal settings, the Cu-Sn/CP electrode displayed wide linear ranges of 0.0005–2.0 mM and 2.0–10.0 mM, respectively. The LOD was as low as 0.061 µM (S/N = 3). The Cu-Sn/CP electrode also displayed exceptional selectivity and stability. Moreover, the developed glucose sensor is proven to be apt for the detection of glucose in human serum samples [52]. It is important to compare the activities such as wide linear range, sensitivity and the LOD of a recently reported bimetallic or bimetallic alloy nanomaterial based glucose sensor as shown in Table 1.

## 3. Bimetallic Nanocomposite Materials Based Glucose Detection

Graphene and carbon nanotubes (CNTs) are potential carbon material analogs; owing to their ease functionalization, notable biocompatibility, good conductivity, and the large specific surface area, they have been extensively explored in the electrode transducer materials [63,64]. Nanocomposites prepared from graphene, carbon nanotube and polymer matrix can not only reduce the interior resistances of electrode but also inhibit the stacking of graphene layers, and then decrease the adsorption of carbon nanotubes [63,65]. The Mu research group prepared a various aspect ratio of Au and Ag nanoparticles electro-deposited on the reduced graphene oxide (RGO)/glassy carbon (GC) surface to form Au/Ag/RGO/GC and Ag/Au/RGO/GC electrodes. The electro-deposition potential of Au or Ag was set at −0.30 V (vs. SCE). The electro-chemical oxidation of glucose was performed in alkaline NaOH solutions. The experimental results displayed that Ag in the bimetallic Au-Ag electrodes plays an important role in the electro-catalytic glucose oxidation, i.e., the glucose oxidation response was highly sensitive to the ratio of Ag to Au loading density in the bimetallic components. The glucose electro-oxidation current density of the bimetallic Au-Ag electrodes was about 0.24 V, which is 4.5 times as large as the Au nanoparticles deposited on the GC electrode. The obtained improved current density was mainly attributed to the synergetic catalytic effect of Au and Ag metals. Notably, the Ag/Au/RGO/GC electrode lost only 26.2% of its initial activity after 500 cycles in 10.0 mM glucose + 0.10 M NaOH solution, which is better than that of the Au/Ag/RGO/GC electrode, Ag/Au/GC, and Au/Ag/GC electrodes [66].

Later, Li et al., synthesized a series of bimetallic MCo (M = Cu, Fe, Ni, and Mn) nanoparticles surrounded in carbon nanofibers (CFs) through electro-spinning and following thermal treatment method [67]. The electrochemical responses for non-enzymatic glucose detection were assessed by CV and chronoamperometry. The findings exhibited that the catalytic capabilities followed a given order of as-synthesized bimetallic nanoparticles, CuCo–CFs > FeCo–CFs > NiCo–CFs > Co–CFs > MnCo–CFs (Figure 4A). It was well known that an advantage from the 3D network films and the synergistic effect of the Co(III)/Co(IV) and Cu(II)/Cu(III) redox couples, CuCo–CFs showed superior detection efficacy even for the glucose detection in human serum samples (507 μA cm^−2^ mM^−1^, the LOD of 1.0 µM (S/N = 3), detection time within 2 s, a linear range from 0.02 to 11 mM, excellent reproducibility, long-term stability, and anti-interference to electroactive molecules or Cl^−^) [67]. Deng et al., reported an NiFe alloy nanoparticle/graphene oxide hybrid (NiFe/GO) for electrochemical sensing of glucose. The CV and amperometric measurements showed that the NiFe/GO-modified GC electrode showed higher sensitivity of 173 µA mM^−1^ cm^−2^ and an LOD of 9 µM (S/N = 3) for glucose sensing with a linear range up to 5 mM, which is better than that of conventional Ni nanoparticles. Moreover, high selectivity for glucose detection was also achieved by the NiFe/GO hybrid [64].

Recently, Lakhdari et al., fabricated an NiFe-Polyaniline electrode via the oxidation of the monomer aniline on FTO (fluorine tin oxide) substrate, followed by deposition of nickel–iron nanoparticles (NiFe NPs) on the polyaniline surface by a chronoamperometry method. The NiFe-PANi nanohybrid electrode exhibited excellent sensing performances, such as sensitivity (1050 μA mM^−1^ cm^−2^), extensive linear range (from 10 μM^−1^ mM), and the LOD of 0.5 μM (S/N = 3) [65]. Zhuang et al., made a nanocomposite of Cu/Cu_2_O@C material, which had a Cu/Cu_2_O hetero-junction with a “head to head” mode homogeneously diffused in the octahedral carbon framework (Cu-MOF) under a moderately mild temperature. The Cu/Cu_2_O@C nanocomposite was then used to construct a glucose sensor (Figure 4B), and showed tremendous sensing responses with high detection sensitivities of 621.12 μA mM^−^^1^ cm^−^^2^ and 372.67 μA mM^−^^1^ cm^−^^2^ toward different glucose concentration ranges of 0.001–1.7 mM and 1.7–9.7 mM, respectively. The displayed sensing results can be attributed to the distinctive features of Cu/Cu_2_O@C: (i) the highly dispersed Cu/Cu_2_O nanoparticles initiating from a metal-organic framework deliver copious catalytic-sites for the glucose oxidation; (ii) the “head to head” mode for Cu/Cu_2_O hetero-junctions allow Cu and Cu_2_O to completely contribute in the electro-catalytic oxidation for glucose; and (iii) an in-situ produced carbon framework supports the charge transfer in the electro-catalytic process [53]. Interestingly, Li et al. developed a highly sensitive electrochemical dual signal glucose sensor in the presence of dopamine (DA), uric acid (UA) using nanocomposites derived from copper and cerium bimetallic nanoparticles, and carbon nanomaterials of graphene and single-walled carbon nanotubes (Figure 4C) [68]. Notably, in the co-occurrence system of DA, UA, and glucose, three well-defined peaks were displayed in CV and DPV measurements. The linearity ranges were obtained i.e., 0.1–100 mM for DA, 0.08–100 mM for UA, and 1–1000 mM for glucose with DPV, and the LOD was found to be 0.0072 μM, 0.0063 μM, and 0.095 μM for DA, UA, and glucose, respectively. Furthermore, the developed system was effectively used to the detect the concentration of DA, UA, and glucose in blood serum samples [68]. It is important to compare the activities such as wide linear range, sensitivity, and LOD of recently reported bimetallic nanocomposite materials based glucose sensors as shown in Table 2.

## 4. Metal-Organic-Framework Derived Bimetallic Nanomaterials Based Glucose Detection

Metal-organic frameworks (MOFs), a class of porous crystalline materials composed of metal nodes and organic linkers, are of distinct attention as catalysts for glucose detection, owing to their numerous active sites, tunable morphologies and various composition [80,81,82]. Precise detection of glucose is vital for the identification of diabetes, where efficient and sensitive biosensors for glucose detection are highly desired. Xia and co-workers developed a high-performance enzyme cascade based biosensing platform by linking MOFs-based nanozyme and natural enzymes. Initially, a porous mixed bi-metal oxide (MnCo_2_O_4_) @ MOF with nanorod-like structures was prepared. Secondly, the nanozyme of bovine serum albumin-Pt nanoparticles integrated with mesoporous MnCo_2_O_4_ (BSA-PtNP@MnCo_2_O_4_) was obtained and applied to build an enzyme cascade biosensing platform. Owing to the synergistic effect of protein, bimetal oxide, and PtNPs, the nanozyme offered tremendous dual enzyme activity. Notably, BSA-PtNP@MnCo_2_O_4_ was utilized as a nanozyme with oxidase activity to attain improved detection of glutathione with an LOD of 0.42 μM. Furthermore, BSA-PtNP@MnCo_2_O_4_ was also used as nanozyme with pronounced peroxidase activity and as a support for functionalization of glucose oxidase (GOx), managing an ordered highly-efficient enzyme based cascade biosending glucose detection platform. The developed glucose biosensing platform has collective benefits of nanozyme and natural enzyme, and delivered excellent glucose detection with the detection limit of 8.1 μM (Figure 5A). The tandem catalytic system not only broadened the application of nanozyme in natural enzyme catalysis, but also provided a simple, efficient, and organized enzyme cascade bio-platform for biosensing and other applications [83].

Next, Wang et al., fabricated a glucose biosensor based on field-effect-transistor (FET) with bimetallic nickel/copper metal-organic frameworks (Ni/Cu-MOFs) as its network layers, which were grown-up by in situ through a facile one-step hydrothermal method, and the sensor electrode was linked with glucose oxidase (GOD) with help of glutaraldehyde (GA) as linkers. Owing to the synergistic effect of Ni ions and Cu ions in MOFs, the biosensor (GOD-GA-Ni/Cu-MOFs-FET) electrode displayed a good field effect response toward glucose via enzymatic reactions. The developed glucose biosensor electrode displayed a wide range (1 μM–20 mM), and a high sensitivity (26.05 μAcm^−2^ mM^−1^) in the lower concentration (1–100 μM) and an LOD of 0.51 μM (Figure 5B). Furthermore, the fabricated biosensor also had some key advantages such as high specificity, excellent reproducibility, good short-term stability, and fast response time [84]. It is important to mention here that enzymatic electrochemical glucose sensors are mainly based on two key enzymes, glucose oxidase (GOx), and glucose dehydrogenase (GDH), which are typically selective and sensitive [85]. However, their inherent disadvantages, such as vulnerability to environmental parameters that affect their response like temperature, pH, and toxic contaminants, and also high fabrication cost, poor reproducibility, and complexed enzyme immobilization stages, have restricted their capability in practical applications. Hence, there is a high demand for the development of non-enzymatic glucose sensors with lower cost, higher sensitivity, faster responsive time, and longer stability.

Recently, the Hou research group prepared nickel/cobalt (NiCo) alloy nano-particles functionalized in graphitized carbon by pyrolyzing a bimetallic (Ni and Co) metal organic framework (NiCo-MOF) at 800 °C under a N_2_ atmosphere (called as NiCo/C) (Figure 5C) [86]. The non-enzymatic glucose sensor comprising SPE/NiCo/C (SPE = screen-printed electrode) showed a high sensitivity of 265.53 μA·mM^−1^·cm^−2^, with an LOD of 0.2 μM (S/N = 3). On the other side, the NiCo/C sensor showed good selectivity for the amperometric detection of glucose. Furthermore, glucose levels were detected with acceptable reliability and accuracy in human real serum samples. Later, Ding et al. fabricated a novel-type of electrochemical detection method for glucose using CuOx@Co_3_O_4_ core-shell nanowires on Cu foam substrate as a potential electrode, which was obtained by stepwise preparation, including anodized nano-sized Cu(OH)_2_ wires, MOFs-wrapped Cu(OH)_2_ nanowires, and finally a calcination step. An as-fabricated hierarchical composite-MOF electrode exhibited the structural characteristics of CuOx nanowires as core and Co_3_O_4_ nanoparticles as a shell that calcinated from using microporous ZIF-67. The constructed glucose sensor was highly comparable to pure mono-metallic oxides, and exhibited higher sensitivity (27,778 μA mM^−1^ cm^−2^ in the range from 0.1 to 1300.0 μM), an LOD of 36 nM (S/N = 3), and faster response time (~1 s) [87]. Furthermore, it has also displayed acceptable selectivity, reproducibility, and longer storage stability. In the meantime, it attained potential results of glucose detection in the real human blood serum sample, which compared to commercial sensors. In addition, four other types of self-supporting MOF-bimetallic oxides core-shell nanowire arrays on Cu foam were also fabricated by using an identical three-step protocol, including CuOx@Fe_2_O_3_, CuOx@NiO, CuOx@CuOx, and CuOx@ZnO core-shell nanowires, indicating the flexibility of this method. These results showed an as-synthesized CuOx@Co_3_O_4_ MOF-bimetallic oxide based glucose sensor and potential usage in the progress of enzyme-free glucose monitoring. Xu et al. reported the enzyme-free glucose detection using metal organic framework (MOF) derived bimetallic Ni/Co nanorods modified with carbon cloth electrodes. The MOF based bimetallic Ni/Co nanorods were synthesized through a facile and simple hydrothermal route, and the synergic catalytic effect of Ni and Co elements; the Ni/Co-MOF(HHTP)/CC not only delivers larger surface area and large effective active sites, but also increases the charge transports and electro-catalytic performance [81]. Under optimized conditions, the Ni/Co-MOF(HHTP)/CC displays tremendous activity with a wide linear range of glucose from 0.3 μM to 2.3 mM, a faster electrode detection response time of 2 sec, a lowest detection limit of 100 nM (S/N = 3), and a high sensitivity of 3250 μA mM^−1^ cm^−2^. Furthermore, the Ni/Co-MOF(HHTP)/CC was effectively applied for the detection of glucose in real samples such as serum and beverages with reasonable detection responses. This facile and economical method provided a novel approach for the screening of glucose in biological and food samples [81]. It is important to compare the activities such as wide linear range, sensitivity, and LOD of recently reported bimetallic MOF nanomaterials based glucose sensors as shown in Table 3.

## 5. Translation Aspects Based Glucose Detection

It is well known that single-use glucose biosensor strips were customarily used for the day-to-day screening of glucose level and then diabetes testing. An ideal diabetic nanodevice for the constant screening and quick- and precise-response detection of blood sugar level is still a task that needs to be completed. Up to now, profitable diabetic sensors and invasive and non-invasive screening systems have been used for early-stage glucose detection and diabetic care. Invasive glucose screening delivered a direct detection of blood glucose level by using simple and easy devices. This device-related analysis was based on needle pinching to release blood in the bar or saliva to be examined [93,94]. Many patients still struggle to use glucometers with needles. Hence, an alternate, non-invasive technique is required to monitor glucose levels. For instance, physiological samples, including interstitial fluid (ISF), tears, saliva, and sweat specimens were used to detect blood glucose concentrations [95,96].

Electroplating of nanoporous Pt (nPt) provides a very strong flexible stress, which results in the exfoliation of nPt on a stretchy polymer substrate in spite of the plasma process to increase adhesion. Yoon et al. established a wearable, robust, elastic, and non-enzymatic continuous glucose monitoring method. The stretchy stainless-steel was highly effective in enhancing the adhesion between the metallic layer and substrate. This wireless method included electrochemical analysis circuits, a micro-controller unit, and a wireless communication module. Lastly, two animal tests were applied using a continuous glucose monitoring method by embedding into subcutaneous tissue and measuring interstitial fluid (ISF) glucose values at 5–15-min time periods. Comparison of the measured ISF glucose with blood glucose determined by the Clarke error grid analysis showed that 82.76% of the measured glucose was within zone A [97]. Arakawa and co-workers developed a cellulose acetate (CA) membrane as an interference rejection membrane on a glucose sensor to measure glucose in saliva. Glucose is effectively measured in vivo directly in human saliva. A mouthguard (MG) glucose sensor is established to screen salivary glucose that is described to be associated with the blood glucose level. Salivary components of ascorbic acid (AA) and uric acid (UA) hinder the accurate measurement of the glucose concentration of human saliva. Thus, CA-coated electrodes were fabricated to study the interference rejection membrane. To detect hydrogen peroxide, a reaction product of glucose oxidase, the effects of AA and UA were also examined. Features of the fabricated biosensor were examined on the basis of artificial saliva. The as-prepared MG sensor can measure the glucose concentration in the range of 1.75−10,000 μmol/L, which comprises a salivary sugar concentration of 20−200 μmol/L (Figure 6) [94]. For the measurement of saliva samples collected from healthy subjects, the output corresponding to the concentration is confirmed; this suggests the possibility of glucose measurement. This MG glucose system can be considered as a suitable route for the open and non-invasive screening of saliva glucose for the management of diabetes patients.

Janyasupab et al. demonstrated a comparative electrochemical study of cobalt/iron (Co-Fe) catalyst on N-doped graphene (NG) for non-enzymatic glucose detection, carried out in physiological pH urine comprising (i) modified artificial urine medium (mAUM), (ii) standard urine (Surine), and (iii) human urine specimens. With no requirement of strong alkaline addition, catalytic properties of Co-Fe-NG were evaluated by using CV and DPV on a GCRDE electrode. Upon consecutive addition of glucose from 0 to 3 mM, DPV results showed two anodic peaks at +0.18 V and +0.42 V vs. Ag/AgCl, corresponding to Co^3+^ and Co^4+^ as a result of glucose binding in urine. By assessing at +0.18 V, the sensitivities of Co-Fe-NG were expected to be 16.77 (R^2^ = 0.987), 45.36 (R^2^ = 0.988), and 20.26 (R^2^ = 0.991) μAmM^−1^ cm^−2^ with the limit of detection of 0.25, 0.07, and 0.19 mM in mAUM, Surine, and human urine specimen with low serum creatinine, respectively [98]. Furthermore, the properties of Co-Fe on graphene (G) and carbon Vulcan XC-72 (C) were also considered in the comparison of NG on the bimetal. Interestingly, Co-Fe-C displayed a good electrochemical trend in glucose detection in urine. However, a negligible catalytic activity was obtainable in Co-Fe-G. Thus, electrochemical responses of Co-Fe-C were also further studied in the comparison of Co-Fe-NG in each type of urine.

## 6. Conclusions and Perspectives

This present review covered the selective reports on the fabrication of enzymatic and non-enzymatic glucose sensors based on the bimetallic nanostructured systems. In the bimetallic nanostructured enzyme based glucose sensors, only glucose oxidase (GOx) was discussed because it has been the most used and highly beneficial enzyme so far. In contrast, various non-enzymatic methods representing the fabrication schemes, characterization systems, wide linear working ranges, LOD, and analytical outcomes were presented in detail. It is important that some non-enzymatic approaches based on bimetallic nanostructures have been expanding their role tremendously. Researchers are making their effort to explore each and every possible transformation to integrate several combinations of two or more nanostructures in order to attain the high sensitivity and selectivity, etc.

Some opportunities and substantial developments are expected in the near future. For instance, commercialization of sensors is the primary condition. The sensors required being single-use, cheaper, and simple to assemble. One of the vital concerns is to make disposable/reusable glucometers with superior exactitude and long shelf-life that can perhaps lead to connecting all the features into elegantly planned microfluidic chips. The existing glucose sensor marketplace has actually minor input in a continuous-glucose-monitoring-system (CGMS) and wearable and implantable equipment. The implantable glucose equipment has the most distinctive features and showed the aptitude to lead the current form of diagnosis and treatments while understanding the practical method towards bulk-production of glucose sensors by exploiting the nanomaterial, which can offer high performance in terms of sensitivity, faster responsive time, and LOD. The synthesis and application of non-toxic nanomaterial is vital for an implantable glucose sensor. It can be worth stating that, regardless of the allied health threats, nanostructured glucose sensors delivered numerous useful features that are abundant for the future inspiration to progress and transform glucose sensing research.

## Figures and Tables

**Figure 1 micromachines-13-00076-f001:**
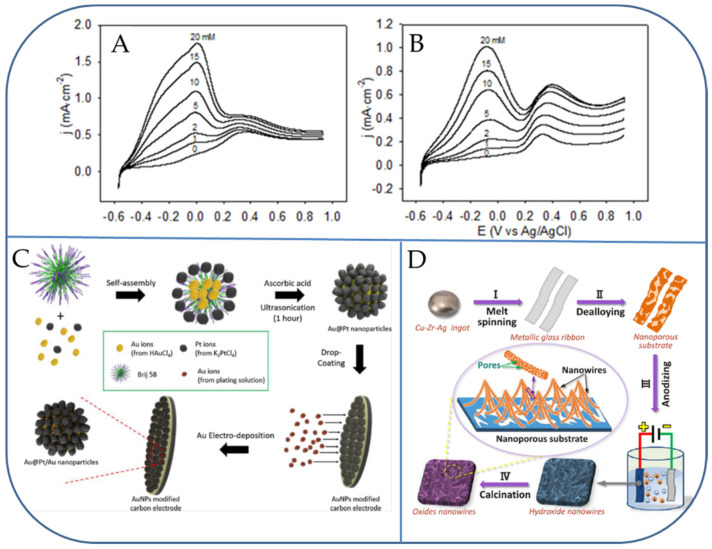
CVs of nanoporous Pt-Pb (50%) electrode at a scan rate of 10 mV s^−1^ in 0.1 M phosphate buffer (pH 7.4) solution containing glucose (at concentrations of 0, 1, 2, 5, 10, 15, 20 mM) and (**A**) in the absence of NaCl and (**B**) in the presence of 0.15 M NaCl. Schematic display showing the fabrication of core-shell structured Au@Pt Au NPs (**C**), and free-standing Cu_x_O/Ag_2_O@NP-CuAg bimetallic electrode (**D**) for glucose detection. Reproduced with permission from Refs. [45,48,49].

**Figure 2 micromachines-13-00076-f002:**
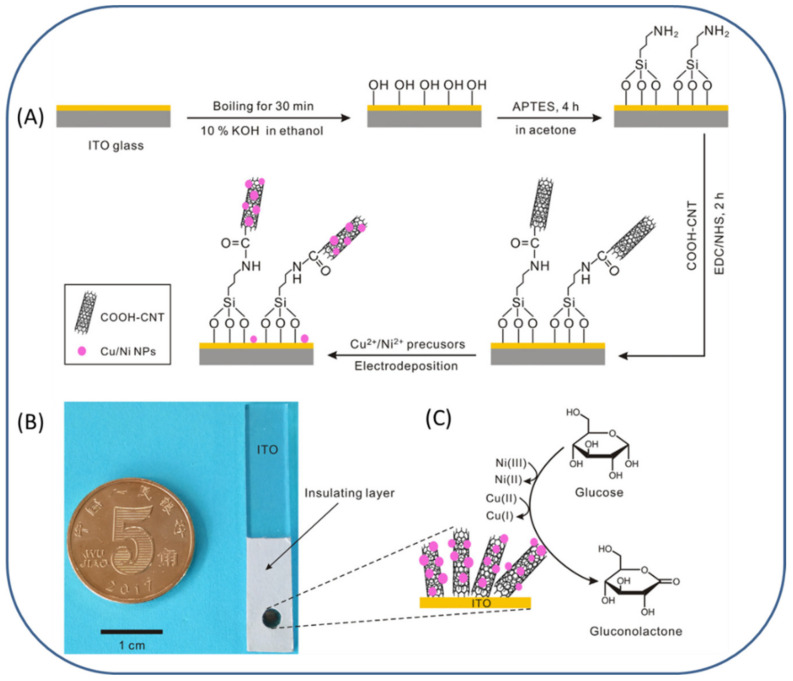
(**A**) Schematic representation showing the preparation of a Cu/Ni NPs CMWCNTs catalyst at the ITO substrate; (**B**) photograph of the as-prepared glucose electrochemical sensor. The total size of the sensor was 4 cm-length × 0.7 cm-width, revealing a 3 mm-diameter area for nanocatalyst assembly; (**C**) schematic illustration of the electrochemical reaction of glucose on the interface of Cu/Ni-CMWCNTs composite. Reproduced with permission from Ref. [50].

**Figure 3 micromachines-13-00076-f003:**
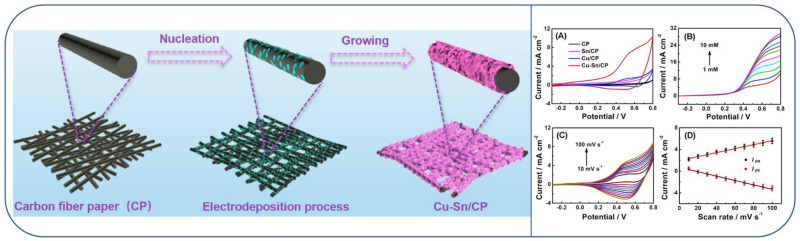
(Left): Schematic fabrication of Cu-Sn alloy nanosheet arrays. (Right): (**A**) CVs obtained for CP, Sn/CP, Cu/CP, and Cu-Sn/CP electrodes in the presence of 1 mM glucose in 0.1 M NaOH at a scan rate of 100 mV s^−1^; (**B**) LSVs obtained for Cu-Sn/CP electrode using various concentrations of glucose in 0.1 M NaOH at a scan rate of 100 mV s^−1^; (**C**) CVs obtained for Cu-Sn/CP electrode in the presence of 1 mM glucose in 0.1 M NaOH at various scanning rates; (**D**) plot obtained used as the function of glucose oxidation peak currents vs. the various scan rates. Reproduced with permission from Ref. [52].

**Figure 4 micromachines-13-00076-f004:**
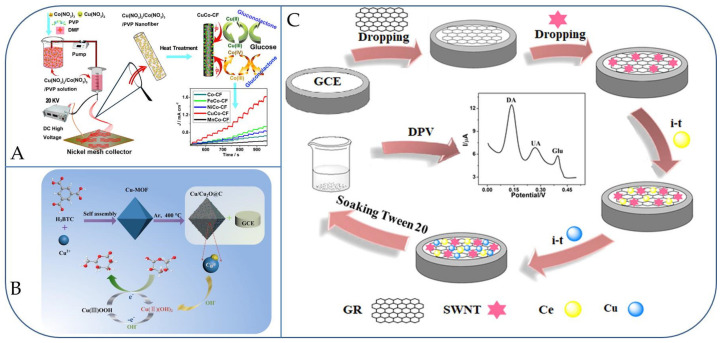
(**A**) Diagrams illustrating the fabrication of CuCo–CFs; (**B**) Cu/Cu_2_O@C/GCE; and (**C**) GR-SWCNT-Ce-Cu-Tween 20/GCE nanostructured electrodes for electro-chemical oxidation of glucose. Reproduced with permission from Refs. [53,67,68].

**Figure 5 micromachines-13-00076-f005:**
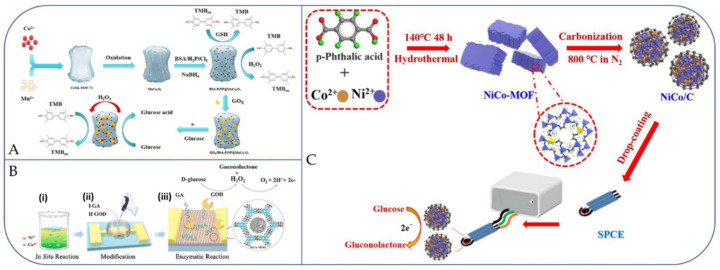
(**A**) Schematic fabrication and biosensing mechanism of BSA-PtNP@MnCo_2_O_4_-based highly-efficient enzyme cascade glucose biosensing platform; (**B**) schematic fabrication of GOD-GA-Ni/Cu-MOFs-FET electrode for glucose biosensing. (Step “i”) The FET was suspended on the surface of reaction solution with mixed bimetallic ions. (Step “ii”) The Ni/Cu-MOFs was developed by dipping GA and GOD successively for enzymatic reaction. (Step “iii”) Ions produced from the enzymatic reaction of glucose collected on the surface of bimetallic MOFs surface, and the oxidized products of glucose; (**C**) the fabrication steps of NiCo/C electrode for nonenzymatic glucose biosensing. Reproduced with permission from Refs. [83,84,86].

**Figure 6 micromachines-13-00076-f006:**
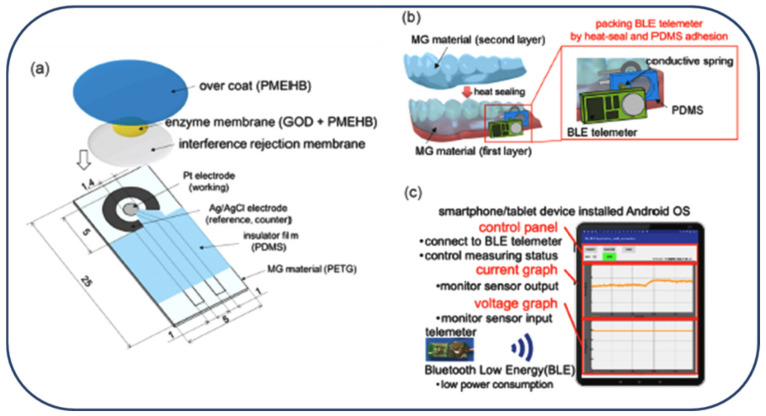
Pictorial display of whole MG glucose sensor and measurement system. (**a**) Electrode fabrication for glucose sensor; (**b**) mounting of MG-type sensor using a conductive spring and PDMS sheet; (**c**) portable wireless detection system using Android tablet devices. Reproduced with permission from Ref. [94].

**Table 1 micromachines-13-00076-t001:** Bimetallic or Bimetallic alloy nanomaterials based glucose sensor fabrication schemes and their analytical performances.

Bimetallic Alloy Nanomaterials	Linear Range Concentration of Glucose	Sensitivity	Detection Methods	LOD	References
				(µM)	
Cu-Sn alloy nanosheets arrays	0.005 mM–2 mM and 2 mM–10 mM	-	CV and CA	0.061	[52]
Ni-Cu bimetallic alloy nanoparticles	0.1–30 µM	1754.7 μA mM^−1^ cm^−2^	CV and CA	0.005	[54]
Ni-Pt nanosheets arrays	1 µM–10.8 mM	2225.5 μA mM^−1^ cm^−2^	CV and CA	0.40	[55]
Ni-Au bimetallic nanostructures	5 µM–3.5 mM and 3.5 mM–7 mM	893.0 μA mM^−1^ cm^−2^	CV and CA	0.70	[56]
Au-Ni alloy nanoparticles	1 µM–1.7 mM	1955.0 μA μM^−1^ cm^−2^	CV and CA	0.41	[57]
CoxP/NiCo-LDH heteronanosheet arrays	1 µM–3 mM	5732.1 μA mM^−1^ cm^−2^	CV and CA	0.60	[58]
Pd_x_Cu_y_ alloy nanoparticles	-	10.1 mA mM^−1^ cm^−2^	CV and CA	10.0	[59]
NiS Nanoclusters@NiS Nanospheres	20 µM–5 mM	54.6 μA mM^−1^ cm^−2^	CV and CA	0.0083	[60]
NiCo/LDH nanoflake	0.5 µM–3000 µM	23,000.0 μA mM^−1^ cm^−2^	CV and CA	0.23	[61]
Cd-In_2_O_4_ nanoparticles	20 µM–1 mM	3.2 mA mM^−1^ cm^−2^	CV and CA	0.08	[62]

LDH: Layered double hydroxide; CV: Cyclic voltammetry; CA: Chronoamperometry.

**Table 2 micromachines-13-00076-t002:** Bimetallic nanocomposite materials based glucose sensor fabrication schemes and their analytical performances.

Bimetallic Nanocomposite Materials	Linear Range Concentration of Glucose	Sensitivity	Detection Methods	LOD	References
				(µM)	
Co_3_O_4_-Ag NWs/Graphene	3 µM–2000 µM	2.49 μA μM^−1^ cm^−2^	CV and CA	0.98	[63]
Ni-Fe@polyaniline	10 µM–1 mM	1050 μA mM^−1^ cm^−2^	CV and CA	0.5	[65]
Ce-Cu/Graphene/SWCNT	1–1000 µM	-	CV and CA	0.095	[68]
CuSn/CNFs Nanocomposites	0.1–9000 µM	-	CV and CA	0.08	[69]
NCNTs/Co-Cu nanocomposites	0.05–2.5 mM	1027 μA mM^−1^ cm^−2^	CV and CA	0.15	[70]
Cu-Ag bimetallic nanocomposites	0.01–30 mM	1340 μA mM^−1^ cm^−2^	CV and CA	0.6	[71]
Cuo encapsulated Ni/Co bimetal Prussian blue	-	600 μA mM^−1^ cm^−2^	CV and CA	0.69	[72]
Cu-Co nanocomposites	1 µM–825 µM	567 μA mM^−1^ cm^−2^	CV and CA	3	[73]
CuO-NiO nanocomposites	0.2 µM–1 mM	4022 μA mM^−1^ cm^−2^	CV and CA	0.08	[74]
Cu-Ni/mesoporous carbon	0.005 mM–0.45 mM	11.7 mA mM^−1^ cm^−2^	CV and CA	5.2	[75]
CoFe Prussian blue	0.1 mM–8.2 mM	18.69 μA mM^−1^ cm^−2^	CV and CA	67	[76]
Cu-Cu_2_O composite nanoparticles	1 µM–7.8 mM	28071 μA mM^−1^ cm^−2^	CV and CA	2.46	[77]
Ag-Ni@MWCNTs	1 µM–4 mM	1485 μA mM^−1^ cm^−2^	CV and CA	7	[78]
Cu-Cu2+1O carbon spheres	0.3 µM–24.5 mM	-	CV and CA	0.06	[79]

SWCNT: Single walled carbon nanotubes; MWCNT: Multi walled carbon nanotubes.

**Table 3 micromachines-13-00076-t003:** Bimetallic MOF nanomaterials based glucose sensor fabrication schemes and their analytical performances.

Bimetallic MOF Nanomaterials	Linear Range Concentration of Glucose	Sensitivity	Detection Methods	LOD	References
				(µM)	
Ni-Co/MOF	0.32 μM–2.32 mM	3250 μA mM^−1^ cm^−2^	CV and CA	0.1	[81]
NiCo/MOF@NiCo_2_O_4_	0.001 mM–7.8 mM	-	CV and CA	0.48	[82]
Ni-Cu/MOF	1 μM–20 mM and 1 μM–100 μM	26.05 μA mM^−1^ cm^−2^	CV and CA	0.51	[84]
CC@MOF-74(NiO)@NiCo LDH	10 μM–1.1 mM and 1.5–9 mM	1699 μA mM^−1^ cm^−2^	CV and CA	0.278	[88]
CuCo-MOF/NF	-	8304.4 μA mM^−1^ cm^−2^	CV and CA	23	[89]
Co-Zn MOF	0.001–0.255 mM and 0.255–2.53 mM	1218 μA mM^−1^ cm^−2^	CV and CA	4.7	[90]
Co-MOF@AgNPs	33 μM–400 μM	13.014 μA μM^−1^ cm^−2^	CV and CA	0.49	[91]
CuCo-MOF	-	6.861 mA mM^−1^ cm^−2^	CV and CA	0.12	[92]

MOF: Metal organic framework; NF: Nickel foam.

## Data Availability

Not applicable.

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
