# Peer review of "Bimetallic Nanomaterials-Based Electrochemical Biosensor Platforms for Clinical Applications"

_micromachines, 2021, doi:10.3390/mi13010076_

Round 1
Reviewer 1 Report
This manuscript reviewed various types of bimetallic electrochemical biosensors, but it is not presented in an organized way, many contents are not placed in a proper order to show the flow of logic and scientific thinking. Some labels in figures need to be modified. Overall, there are lots of problems in this manuscript that make it very difficult to read. I suggest the authors to rewrite Abstract, and divide the large paragraphs into smaller ones, so that each paragraph can clearly explain one topic. Some details of these problems are listed below, but almost every paragraph has wording and other issues, and they all need to be re-organized.
- This manuscript focuses on glucose biosensing, so the Abstract should focus on it, not just the last sentence.
- In the Introduction section, there are lots of wording problems, and many contents in this section are not related to the contents in later sections. On the other hand, many contents in later sections should be moved to this section.
- In section 2, each paragraph needs to be divided into smaller ones.
- From line 110 to 152, the authors used lots of words to explain one work, then why not adding a figure to show the data in a clearer way. The words used for this work is unproportionally too many compared to other research work discussed in this manuscript, any reason for that or just poorly organized?
- From line 220 to line 223, where did the authors get these equations? Please add reference. Equation iv seems to have balancing problem, so did the authors really mean they are equations?
- The labels in Figure 3, 4, and 5 are not clear.
- In Table 1, 2, and 3, are those LOD values from clinical samples? What solvent or pre-treatment did each work used?
Reviewer 2 Report
Authors have presented a comprehensive review on Bimetallic Nanomaterials-Based Electrochemical Biosensor Platforms for Clinical Applications. They reviewed previously reported sensors, and their limitations. I request to add a separate section covering the Translational Aspect of the previously developed sensors. I also request to add the following references to support the literature of electrochemical biosensors specifically detecting the ascorbic acid level in clinical patients. https://pubs.acs.org/doi/abs/10.1021/acsami.7b01675
Table 1 and 2 show some examples. However, authors are requested to add more details on their technique (electrical/optical/etc) used in detecting the analyte.
Round 2
Reviewer 2 Report
In this revision, authors have added new sections to cover translation aspect (Section 5) and also included new details to show the relevant detecting techniques (Table 3). This review can now be accepted in the present form.